# Digit Span Tests Are More Sensitive than SDMT for Detecting Working Memory Impairment and Correlate with Metabolic Alterations in White Matter and Deep Gray Matter Nuclei in Multiple Sclerosis: A GABA-Edited Magnetic Resonance Spectroscopy Study

**DOI:** 10.3390/ijms26188842

**Published:** 2025-09-11

**Authors:** Ján Grossmann, Marián Grendár, Petra Hnilicová, Nina Kováčiková, Lucia Kotul’ová, Wolfgang Bogner, Egon Kurča, Ema Kantorová

**Affiliations:** 1Clinic of Neurology, Jessenius Faculty of Medicine in Martin, Comenius University Bratislava, 03601 Martin, Slovakia; jangrossmann@gmail.com (J.G.); nina.kovacikov@gmail.com (N.K.); egon.kurca@uniba.sk (E.K.); 2Biomedical Centre, Jessenius Faculty of Medicine in Martin, Comenius University Bratislava, 03601 Martin, Slovakia; marian.grendar@uniba.sk (M.G.); petra.hnilicova@uniba.sk (P.H.); lucia.kotulova@uniba.sk (L.K.); 3Department of Biomedical Imaging and Image-Guided Therapy, Medical University of Vienna, 1090 Vienna, Austria; wolfgang.bogner@meduniwien.ac.at

**Keywords:** SDMT, Digit Span Forward, Digit Span Backward, GABA-edited 1H-MRS, multiple sclerosis

## Abstract

In this paper, we aimed to evaluate the efficacy and usefulness of three brief, easy-to-administer, and repeatable tests, namely SDMT, Digit Span Forward (DSF), and Digit Span Backward (DSB) in MS patients (MSp), and compared the results with those of healthy volunteers (CONs). We were hoping to identify the most sensitive test that could be used regularly in clinical practice. In addition, we tried to identify the metabolic background of the cognitive setting using the advanced radiological method, Mescher–Garwood (MEGA)-edited 1H Magnetic Resonance Spectroscopy (1H-MRS). A total of 22 relapsing MSp and 22 CONs were enrolled. The SDMT, DSF, and DSB tests were used on all participants. The patients also underwent a 1H-MRS brain examination. In addition to N-Acetyl-Aspartate (tNAA), Myoinositol (mIns), Choline (tCho), and Creatine (tCr) were also evaluated GABA and Glutamate–Glutamine (Glx) ratios. CONs were superior to MSp in the results of all neurocognitive tests. The DSB was found to be the most sensitive test for identifying MSp. The SDMT in MSp correlated with inflammatory and degenerative metabolites in the thalamus, hippocampus, and corpus callosum. A correlation between increased Glx- and GABA-ratios and SDMT was found. Unlike the SDMT, the DSF and DSB showed correlations with inflammatory metabolites in the caudate nucleus and hypothalamus. DSF correlated with GABA ratios in the hippocampus. Our study confirms the efficacy of DSF and DSB tests in evaluating working memory cognitive impairment in MSp, showing an association of the tests with specific brain metabolites.

## 1. Introduction

Multiple sclerosis (MS), a chronic, immune-mediated neurodegenerative disorder of the central nervous system (CNS), collectively contributes to progressive neurological disabilities [1,2]. While motor and sensory deficits are hallmark clinical features, cognitive dysfunction represents a profound burden, significantly impairing quality of life, vocational capacity, and psychosocial well-being [3]. Cognitive dysfunction may manifest early in the disease course, even without significant physical disability, and msy correlate poorly with conventional magnetic resonance imaging (MRI) markers such as T2 hyperintense lesion load or gadolinium-enhancing activity [4]. This dissociation underscores the limitations of structural imaging in capturing the complex neurobiological substrates of cognitive decline, which are hypothesized to involve microstructural damage, metabolic dysregulation, and network-level disconnection. Emerging evidence suggests that cortical and deep gray matter (GM) atrophy and diffuse white matter (WM) pathology in normal-appearing tissue may drive cognitive impairment. Yet, the molecular mechanisms linking neuroinflammation, neurodegeneration, and neuropsychological deficits remain poorly understood [5,6,7].

One of the advanced MRI methods is 1-proton Magnetic Resonance Spectroscopy (1H-MRS) [8], which detects biochemical alterations in lesioned and normal-appearing brain tissue [4]. In this study, Mescher–Garwood (MEGA)-edited 1H-MRS [9,10] was used, which allows the evaluation of neurotransmitters such as Glutamate with its precursor Glutamine (Glx), and γ-Aminobutyric Acid (GABA), in addition to traditional metabolites reflecting tissue alteration such as N-Acetyl-Aspartate (tNAA), Myoinositol (mIns), and Choline (tCho), and Creatine (tCr)-containing compounds [9,10]. In a summary, tNAA is a marker of neuronal density and mitochondrial function synthesized in neuronal mitochondria; tCr is a compound central to adenosine triphosphate (ATP) buffering and cellular energy homeostasis; tCho is indicative of membrane phospholipid turnover and upregulated in neuroinflammatory states; mIns is a glial-specific osmolyte elevated in astrogliosis and microglial activation; Glx is an excitatory neurotransmitter involved in synaptic signaling and implicated in excitotoxic injury; and GABA has a role as the major inhibitory neurotransmitter referred to as a marker of neuro-plasticity [11,12].

Cognitive function and affective symptoms have been evaluated using a standardized battery of neuropsychological tests and validated self-report measures selected for their established sensitivity to MS-related deficits. The Symbol Digit Modalities Test (SDMT) [13,14,15] is a validated measure of information processing speed and working memory, the primary endpoint for assessing cognitive efficiency in MS. The SDMT emphasizes rapid visual scanning, sustained attention, and mental flexibility, domains disproportionately impaired in MS. It has been identified as the most sensitive cognitive marker of disease progression [14,16,17].

Verbal working memory and attentional capacity could be assessed using the Digit Span subtest from the Wechsler Adult Intelligence Scale (WAIS-IV), administered in both forward (DSF) and backward (DSB) conditions [18]. DSB requires mental manipulation by reversing digit sequences, taxing executive control, and working memory updating. While DSF primarily reflects phonological loop integrity, DSB engages the central executive component of Baddeley’s working memory model, making it sensitive to frontal-subcortical circuit dysfunction common in MS [19].

In this study, we aimed to evaluate the efficacy and usefulness of three brief, easy-to-administer, and repeatable tests, namely SDMT, DSF, and DSB in MS patients, and compare the results with those of healthy volunteers (CONs). We were hoping to identify the most sensitive test that would be used in clinical practice regularly. In addition, we tried to find the metabolic background of the cognitive setting using the advanced radiological method, MEGA-edited 1H-MRS. Selected areas of white and gray matter in the brains of MS patients (MSp) and controls (CONs) were evaluated; data were compared and correlated with the results of the SDMT, DSF, and DSB tests.

## 2. Results

Table 1 shows differences between MSp and CONs in demographic status (e.g., age, EDSS, disease duration, MRI activity, treatment method) and results of cognitive tests (SDMT, DSF, DSB).

The average score of DSF in CONs was 7.86 (0–8) and in MSp 6.08 (0–6). The average score of DSB in CONs was 7.14 (0–9) and in MSp 5.09 (0–10).

The Cochran–Armitage test for trends provided an alternative hypothesis, namely that DSF scores decrease in MSp (Z = 3.2861, dim = 9, *p*-value = 0.0005079). It revealed that as the DSB series increased, the relative proportion of MSp decreased.

The Cochran–Armitage test for trends provided an alternative hypothesis, namely that DSB scores decrease in MSp (Z = 3.9294, dim = 8, *p*-value = 0.00004258). It revealed that as the DSB series increased, the relative proportion of MSp decreased.

### 2.1. Correlations of Brain Metabolites with Cognitive Tests Can Be Found in Table 2, Table 3 and Table 4

Metabolic changes in parts of GM and WM that were independent of the presence of demyelinating lesions were tested and correlated with cognitive tests. Data are in Table 2, Table 3 and Table 4.

### 2.2. Evaluation of the Most Significant Predictors of Multiple Sclerosis

A false positive and false negative rate of approximately 20% should be expected when using the RF machine learning algorithm (trained with the above predictors) for predicting whether a person has MS or is a CON. RF results prediction power was high, and the area under the curve was 86%.

Essentially, the same ranking was obtained by the basic variable importance (VIMP) ranking algorithm via random forest, see Figure 1.

Here is an arrangement of predictors according to RF (we show *p*-values).

DSB series (Fisher’s exact test *p*-value < 0.001, decreased in MSp);Caudate nucleus mIns/tNAA (Wilcoxon’s *p*-value < 0.001; increased in MSp);Caudate nucleus mIns/tCr (Wilcoxon’s *p*-value < 0.001; increase in MSp);Caudate nucleus tCho/tNAA (Wilcoxon’s *p*-value < 0.001; increased in MSp);SDMT (Wilcoxon’s *p*-value < 0.001; decreased in MSp);DSF series (Fisher’s exact test *p*-value = 0.001, decreased in MSp);Corpus callosum mIns/tNAA (Wilcoxon’s *p*-value = 0.001; increased in MSp);Caudate nucleus tNAA/tCr (Wilcoxon’s *p*-value = 0.016; decreased in MSp);Caudate nucleus GABA/tCr (Wilcoxon’s *p*-value = 0.001; decreased in MSp).

## 3. Discussion

Our results confirm that MSp performed significantly worse than healthy controls on neurocognitive tests. The SDMT is currently considered the most sensitive instrument for evaluating processing speed in MS. Impairment in processing speed in MS has been shown to underlie deficits in working memory, executive function, and learning [13,14,15]. However, we also found substantial deficits in working memory and attention in MSp, as assessed by Digit Span tests. The Digit Span task assesses attention, encoding, auditory processing, and working memory capacity. Moreover, DSB was found to be the most sensitive test to identify MSp (See Figure 1 and Figure 2). These results support Beatty’s [20] hypothesis that MSp experience both a generalized difficulty in sustaining concentration and a more specific deficit in attentional resource allocation under multi-task demands.

To explore the neural correlates of cognitive dysfunction, we analyzed brain metabolites linked to performance on different cognitive tests.

### 3.1. SDMT

In our study, lower SDMT scores in GM of MSp correlated with the thalamic neuronal loss (decreased tNAA/tCr ratio reflects diminished neuronal integrity), hippocampal microglial activation (elevated mIns/tNAA was associated with poorer SDMT performance), and hypothalamic glutamate reduction (decreased Glx/tCr correlated with lower SDMT scores).

In the corpus callosum (WM), SDMT impairment was associated with increased choline (tCho/tCr), indicating intensified demyelination, and decreased GABA (GABA/tCr), suggesting reduced inhibitory neurotransmission.

SDMT performance has been previously linked to volumes of the thalamus, cerebellum, putamen, and occipital cortex [21,22]. Although we did not measure thalamic volume directly, metabolite-based findings (i.e., reduced tNAA/tCr) support the association between thalamic integrity and SDMT scores. In early-stage MS, we previously reported correlations between elevated tCho and mIns ratios and lower SDMT scores [23], suggesting that metabolic alterations may anticipate neuronal loss. Additionally, 7 T MRS studies in older versus younger individuals report higher glia-related metabolites (mIns, tCr, tCho), particularly in the hippocampus and thalamus [24]. Our hippocampal mIns/tNAA findings align with those results, supporting a potential role of sustained microglial activity in metabolic dysfunction and neurotoxicity in MS.

Heightened hypothalamic glutamate levels correlated with higher SDMT scores, potentially reflecting increased excitatory neurotransmission to support attention and processing speed via cortico-subcortical pathways.

Higher tCho/tNAA ratios in the corpus callosum corresponded with lower SDMT scores-consistent with increased demyelination and axonal loss [8,25]. Morphometric and diffusion studies have reinforced the role of corpus callosum integrity in cognitive processing speed, including SDMT performance [26]. Moreover, we observed a positive correlation between SDMT and GABA levels in the rostral corpus callosum. GABA is a key inhibitory neurotransmitter crucial for maintaining excitatory-inhibitory balance in cortical circuits. Previous GABA-focused MRS studies in MS did not specifically assess the corpus callosum [27,28,29], highlighting the novelty of our findings.

### 3.2. Digit Span Forward and Digit Span Backward Series

In contrast to SDMT scores, as the DSB and DSF series increase, the relative proportion of MSp decreases. It was challenging to identify MSp among individuals with high DSB and DSF scores. DSF and DSB scores, indicating better working memory, were positively correlated with increased mIns ratios in the caudate nucleus, suggesting microglial activation. Although this association has not previously been reported in MS, parallel findings in Human Immunodeficiency Virus (HIV) patients show that working memory deficits are linked to elevated glial metabolites, including mIns in the caudate and associated white matter networks [30].

The caudate nucleus plays a vital role in strategic planning, goal-directed behavior, and both visual and verbal working memory via cortico-subcortical circuits [31,32,33,34]. Functional MRI studies show either caudate nucleus or hippocampus activation specifically during spatial working memory tasks [31,32]. Conversely, SDMT performance relies primarily on information processing speed and engages frontoparietal, visual-attention, and cerebellar networks, with minimal basal ganglia involvement [26].

In our cohort, higher DSF scores correlated with increased hippocampal GABA levels. Preclinical and clinical studies indicate that reduced GABAergic inhibition impairs memory and attention [35] and that lower GABA levels in the prefrontal cortex are linked to greater working memory decline under cognitive load [36]. Moreover, reduced hippocampal GABA has been observed in MS patients [27,28] and is associated with cognitive and motor deficits [23,37].

The GABAergic system is critical for synchronizing large neuronal populations. Inhibitory projections, such as those connecting the medial septum, hippocampus, and entorhinal cortex, are crucial for modulating oscillatory activity underlying episodic memory formation [38].

Additionally, both DSF and DSB scores were negatively correlated with hypothalamic tNAA/tCr and positively with tCho/tNAA, indicating neuronal loss and demyelination. Although comparable data are scarce, these findings align with the known role of the hypothalamus in cortico-subcortical pathways that support learning and motivated behavior [39].

Finally, our data extend findings by Llufriu et al. [40], who identified mIns/tNAA in normal-appearing white matter as predictive of disability in MS. We similarly observed correlations of mIns/tCr and tNAA/tCr in the corpus callosum with reduced DSF and DSB scores, reinforcing the use of these metabolite ratios as biomarkers of cognitive decline.

### 3.3. Limitations of the Study and Suggestions for Further Research

Our study also has several limitations. First, we tested a relatively small number of participants. Second, from a technical perspective, employing absolute metabolite quantification rather than ratio-based analysis could provide more precise biochemical measurements. Additionally, advanced 1H-MRS protocols with higher spatial resolution (7T MRI systems) may improve the detection of localized pathological changes in the brain subregions and also the brain cortex. fMRI studies and evaluation of MRI-tractography could also bring better insights into functional connections between activated brain regions. Correlation with serum biomarkers, such as Neurofilament Light Chain and Glial Fibrillary Acidic Protein, could be another option. These methodological refinements could significantly enhance the future investigations of structure–function relationships in MS-related cognitive pathology.

## 4. Methods

### 4.1. Patients

Patients fulfilling the criteria for definite MS according to McDonald 2017 criteria [2] were randomly selected from the Multiple Sclerosis Centre, University Hospital in Martin, Slovakia, and they were entered into the study after providing written consent. The inclusion criteria were as follows: relapsing course of disease, absence of clinical relapse and corticosteroid treatment within at least three months before study entry, Expanded Disability Status Scale (EDSS) score of ≤5.0, age between 18 and 55 years, and the ability and willingness to cooperate in examinations. The exclusion criteria were severe comorbidities and acute depressive symptomatology confirmed by psychological examination.

We recruited 24 MSp (10 males; 14 females) and 23 CONs (7 males; 16 females), who were age-, education-, and gender-matched. Out of all participants, data from 2 MSp (1 male; 1 female) were not complete, and 1H-MRS data from 1 CON subject (1 female) did not meet the technical quality criteria for evaluation, so these subjects were excluded from the analyses. Finally, 22 MSp (9 males, 40.9%; 13 females, 59.1%) and 22 CONs (7 males; 15 females) participated in the study.

In MSp, clinical disability was evaluated by a neurologist certified in EDSS assessment. Cognitive status was tested using the SDMT, DSF, and DSB from a neuropsychological battery, WAIS-IV, by an experienced neuropsychologist.

The SDMT is based on quickly pairing geometric symbols with corresponding numbers. It was performed orally for 90 s. The DSF and DSB tests were carried out as follows: the participant heard a sequence of digits and then immediately verbally repeated the sequence, either as it was heard (forwards) or in reverse order (backwards). If the participant responded correctly, the subsequent trial presented a more extended sequence. The task terminated when participants responded incorrectly on three occasions within a span length. A participant’s span comprised the longest number of sequential digits that the could accurately remember. Test duration was a maximum of 1 to 3 min. Both MSp and CONs were tested under the same laboratory conditions.

### 4.2. Mescher–Garwood GABA-Edited 1H-MRS

Measurements were performed on a 3 T whole-body MR scanner (TIM Trio^®^, Siemens Healthcare, Erlangen, Germany) using a 32-channel head coil (SiemensHealthcare, Erlangen, Germany) for signal reception. To ensure accurate and reproducible slice positioning, auto-align and three-dimensional T1-weighted anatomical reference images were used (magnetization-prepared two rapid acquisition gradient echoes—MP2RAGE, TR/TE = 4600/3.2 ms, resolution = 0.8 × 0.8 × 0.8 mm^3^). MRS- measurements were performed using a three-dimensional MEGA-edited sequence (TR/TE = 1600/68 ms, 32 averages, with a measurement time of ~20 min) with localized adiabatic spin echo refocusing selection and spiral encoding [9,10]. For real-time correction, volumetric, dual-contrast, echo planar imaging-based navigators that update B0 shim, frequency, and motion artifacts were used. All 1H-MRS slices were placed parallel to the anterior commissure–posterior commissure line, covering the centrum semiovale and basal ganglia, with a volume of interest (VOI) of 90 × 80 × 50 mm^3^, a field of view (FOV) of 160 × 160 × 80 mm^3^, and ~2 cc nominal resolution. The acquired matrix size of 10 × 10 × 10 voxels was interpolated to the 16 × 16 × 16 matrix of multivoxel (CSI, chemical shift imaging) representation.

Required regions were derived based on T1-weighted images, resulting in the selection of areas without the detectable manifestation of focal WM or GM lesions. All MR-spectra were evaluated in LCModel software (Version 6.3–1, S. Provencher, LC Model, Oakville, ON, Canada) for obtaining the following metabolite ratios tNAA/tCr, tCho/tNAA, tCho/tCr, mIns/tNAA, mIns/tCr, Glx/tNAA, Glx/tCr, GABA/tNAA, and GABA/tCr.

For further analytical processes, all selected voxels had to fulfill quality criteria, including a signal-to-noise ratio of over 10 and a full width at half maximum of less than 0.10 ppm.

We chose brain regions that have connections to pathways responsible for cognition and memory. We examined metabolic changes in parts of GM and WM that were independent of the presence of demyelinating lesions.

### 4.3. Statistical Analyses

All statistical analyses were performed in NCSS (version 9.0, LLC, Kaysville, UT, USA). Differences between women and men were not significant. One-way ANOVA was used for statistical analyses that found no differences between metabolic values in the left and right voxels. Differences in demographic and clinical parameters (age, SDMT, DSF, DSB, EDSS) and GABA-edited 1H-MRS metabolite ratios between patients and controls were evaluated using Wilcoxon’s rank-sum test, Fisher’s exact test, or one-way ANOVA with Bonferroni’s post hoc test. The Cochrane–Armitage test was applied to compare the trends of the results of the DSF and DSB series. A random forest (RF) classification algorithm was trained with the default settings of the rfsrc function from the R library RandomForestSRC. Both psychological variables and 1H-MRS metabolite ratios concentrations were used as predictors. The performance of the trained classifier was assessed via the Out-Of-Bag ROC curve and AUC. The predictor ranking was obtained using the variable importance (VIMP) and the graph depth.

## 5. Conclusions

We can conclude that SDMT, DSB, and DSF are the most reliable and sensitive neuropsychological tests, suggesting that they could serve as a day-to-day screening method for neuropsychological impairment in MSp. They are easy-to-perform, non-demanding tests, taking only several minutes. Moreover, the DSB test was found to be the most sensitive in determining MS-related working memory impairments. Additionally, our data show unique metabolic changes in the deep GM and WM of the brain in correlation with cognitive tests. The results confirm the hypothesis that MS is a whole-brain disease.

## Figures and Tables

**Figure 1 ijms-26-08842-f001:**
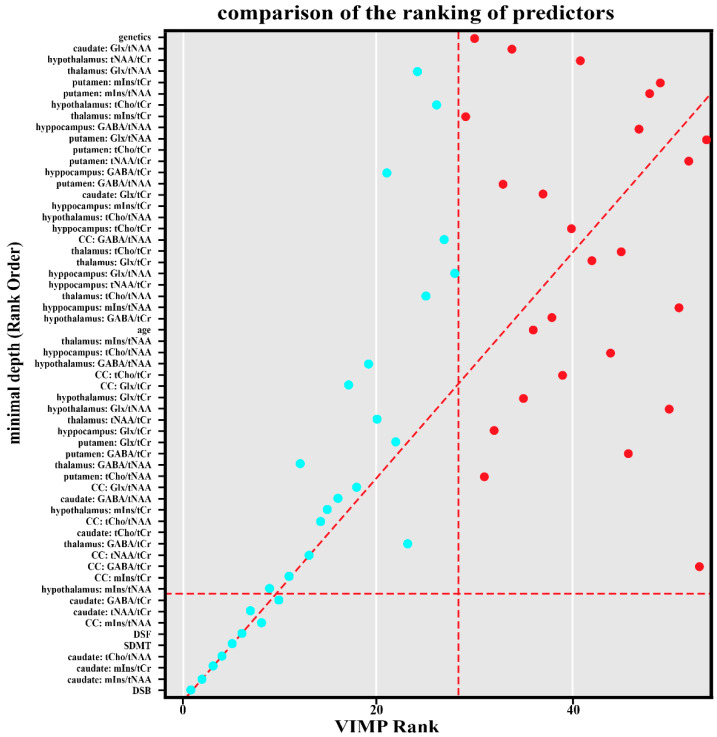
Comparison of the ranking of predictors by graph depth on the y-axis and by VIMP on the x-axis. The horizontal dashed line represents the cutoff for significance within the graph depth. The predictors below the cutoff are considered necessary according to graph depth. Similarly, the vertical dashed line is for cutoff on VIMP. VIMP = variable importance; Backward = Digit Span Backward; caudate_mIns.tNAA = Caudate Nucleus Myoinositol/tN-Acetyl Aspartate; caudate_mIns.tCr = in Caudate Nucleus Myoinositol/t-Creatine; caudate_tCho.tNAA = Caudate Nucleus tCholin/tN-Acetyl Aspartate; SDMT = Single Digit Modality Test; forward = Digit Span Forward; corpus callosum_mIns.tNAA = Corpus Callosum Cyoinositol/tN-acetyl aspartate; caudate_tNAA.tCr = Caudate Nucleus tN-Acetyl Aspartate/tCreatine; caudate_GABA.tCr = Caudate Nucleus γ-Aminobutyric acid/tCreatine.

**Figure 2 ijms-26-08842-f002:**
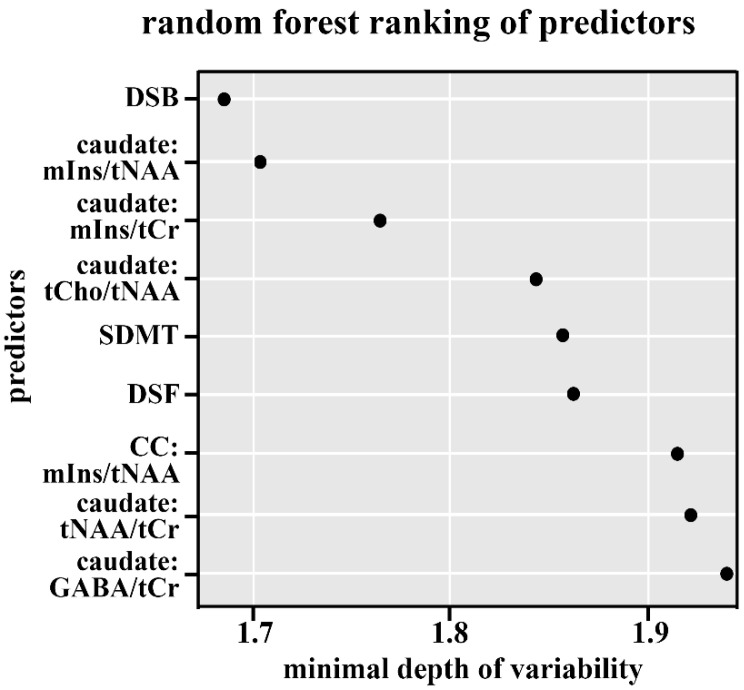
Ranking of predictors obtained by the graph depth method via random forest. On the y-axis, predictors are ranked from the most important at the top to the least important one, which is at the significance cutoff at the bottom. The x-axis represents the graph depth, with smaller values indicating greater importance of the predictor. Backward = Digit Span Backward; caudate_mIns.tNAA = Caudate Nucleus Myoinositol/tN-Acetyl Aspartate; caudate_mIns.tCr = Caudate Nucleus Myoinositol/t-Creatine; caudate_tCho.tNAA = Caudate Nucleus tCholin/tN-Acetyl Aspartate; SDMT = Single Digit Modality Test; forward = Digit Span Forward; corpus callosum_mIns.tNAA = Corpus Callosum Myoinositol/tN-Acetyl Aspartate; caudate_tNAA.tCr = Caudate Nucleus tN-Acetyl Aspartate/tCreatine; caudate_GABA.tCr = Caudate Nucleus γ-Aminobutyric acid/tCreatine.

**Table 1 ijms-26-08842-t001:** Demographic and basic information about the study groups. The table shows differences between MSp and CONs in demographic status (e.g., age, EDSS, ARR, disease duration, MRI activity, treatment) and cognitive tests (SDMT, DSF, DSB).

Characteristic	CON (N = 22; 15F)	MSp (N = 22; 13F)	Statistics: MSp vs. CONs		
			*p*-Value	*q*-Value	Median Diff. (95% CI)
age (years; average, min–max)	30.0 (25.0–34.0)	34.0 (27.0–41.0)	0.2	0.4	−3.0 (−9.0, 2.0)
EDSS (average, min–max)	na	3.28 (1–5)	na	na	na
ARR (years; average)	na	0.25	na	na	na
disease duration (months; average, min–max)	na	76.5 (10–162)	na	na	na
MRI activity (newT2/Gd^+^ lesions)	na	3/22 (13.6%)	na	na	na
Disease-Modifying Treatment	na	NAT: 9/22 (40.9%) DMF: 3/22 (13.6%) GA: 3/22 (13.6%) FIN: 2/22 (9.0%) ALEM: 2/22 (9.0%) TERI: 1/22 (4.5%) none 2/22 (9.0%)	na	na	na
SDMT (average, min–max)	53.5 (47.0–60.5)	43.0 (35.5–49.0)	<0.001	0.003	11 (6.0, 17)
DSF	DSF 3: 0 (0%) DSF 4: 0 (0%) DSF 5: 1 (4.5%) DSF 6: 1 (4.5%) DSF 7: 5 (22.7%) DSF 8: 8 (36.3%) DSF 9: 7 (31.8%) DSF 10: 0 (0%) DSF 11: 0 (0%)	DSF 3: 1 (4.5%) DSF 4: 4 (18.1%) DSF 5: 6 (27.2%) DSF 6: 3 (13.6%) DSF 7: 4 (18.1%) DSF 8: 3 (13.6%) DSF 9: 0 (0%) DSF 10: 1 (4.5%) DSF 11: 1 (4.5%)	0.001	0.010	na
DSB	DSB 3: 0 (0%) DSB 4: 1 (4.5%) DSB 5: 0 (0%) DSB 6: 4 (18.1%) DSB 7: 9 (40.9%) DSB 8: 6 (27.2%) DSB 9: 2 (9.0%) DSB 10: 0 (0%)	DSB 3: 2 (9.0%) DSB 4: 10 (45.4%) DSB 5: 3 (13.6%) DSB 6: 4 (18.1%) DSB 7: 2 (9.0%) DSB 8: 1 (4.5%) DSB 9: 0 (0%) DSB 10: 1 (4.5%)	<0.001	0.003	na

F = female; CONs = controls; MSp = MS patients; EDSS = Expanded Disability Status Scale; ARR = Annualized Relapse Rate; MRI = Magnetic Resonance Imaging; T2 = T2 weighted lesions; Gd+ = gadolinium-enhancing lesions; DSF = Digit Span Forward; DSB = Digit Span Backward; NAT = Natalizumab; DMF = Dimethyl Fumarate; GA = Glatiramer Acetate; FIN = Fingolimod; ALEM = Alemtuzumab; TERI = Teriflunomid; CI = confidence interval; na = not applicable; statistics (*p*-values, *q*-values, and median difference) were evaluated using Wilcoxon rank-sum test; Fisher’s exact test.

**Table 2 ijms-26-08842-t002:** Correlation of SDMT with the 1H-MRS metabolite ratios in the brain areas of MSp.

The Brain Area	1H-MRS Metabolite Ratio	Correlation of SDMT with 1H-MRS Metabolite Ratios	
		*p*-Value	Cor
Thalamus R	tNAA/tCr	0.040	0.430
Hypothalamus L	Glx/tCr	0.032	0.447
Hippocampus L	mIns/tNAA	0.014	−0.507
CC_splenium	tCho/tNAA	0.048	−0.417
CC_rostral	GABA/tCr	0.038	0.436
CC_genu	tCho/tNAA	0.021	−0.479

tNAA = N-acetyl aspartate; Glx = glutamate and glutamine; tCho = choline; GABA = γ-amino butyric acid; mIns = myoinositol; tCr = creatine; tCho = choline; CC = corpus callosum; statistics (*p*-values, correlation coefficient) were evaluated using Welsh two-sample *t*-test, R = right, L = left.

**Table 3 ijms-26-08842-t003:** Correlation of Digit Span Forward series with the 1H-MRS metabolite ratios in the brain areas of MSp.

The Brain Area	1H-MRS Metabolite Ratio	Correlation of Digit Span Forward Series with 1H-MRS Metabolite Ratios	
		*p*-Value	Cor
Caudate R	mIns/tNAA	0.022	0.474
Caudate R	mIns/tCr	0.021	0.477
Hypothalamus R	Glx/tCr	0.010	−0.527
Hypothalamus R	tNAA/tCr	0.040	−0.432
Hypothalamus L	tCho/tNAA	0.012	−0.517
Hippocampus L	GABA/tNAA	0.007	0.545
Hippocampus L	GABA/tCr	0.026	0.464
CC_splenium	mIns/tCr	0.041	−0.429
CC_genu	tNAA/tCr	0.021	−0.479

R = right; L = left; tNAA = N-acetyl aspartate; Glx = glutamate and glutamine; tCho = choline; GABA = γ-amino butyric acid; mIns = myoinositol; tCr = creatine; caudate = nucleus caudatus; CC = corpus callosum; statistics (*p*-values, correlation coefficient) were evaluated using the Welsh two-sample *t*-test.

**Table 4 ijms-26-08842-t004:** Correlation of Digit Span Backward series with the 1H-MRS metabolite ratios in the brain areas of MSp.

The Brain Area	1H-MRS Metabolite Ratio	Correlation of Digit Span Backward Series with 1H-MRS Metabolite Ratios	
		*p*-Value	Cor
Caudate R	tCho/tNAA	0.033	0.446
Caudate R	mIns/tNAA	0.014	0.505
Caudate R	mIns/tCr	0.012	0.517
Hypothalamus R	tNAA/tCr	0.03	−0.453
CC_splenium	tNAA/tCr	0.048	−0.417
CC_splenium	mIns/tCr	0.048	−0.416

R = right, CC = corpus callosum; tNAA = N-acetyl aspartate; tCho = choline; GABA = gaba amino butyric acid; mIns = myoinositol; tCr = creatine; caudate = nucleus caudatus; CC = corpus callosum; statistics (*p*-values, correlation coefficient) were evaluated using Welsh two-sample *t*-test.

## Data Availability

No new data were created.

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
