# Peer review of "Digit Span Tests Are More Sensitive than SDMT for Detecting Working Memory Impairment and Correlate with Metabolic Alterations in White Matter and Deep Gray Matter Nuclei in Multiple Sclerosis: A GABA-Edited Magnetic Resonance Spectroscopy Study"

_ijms, 2025, doi:10.3390/ijms26188842_

Round 1

Reviewer 1 Report

Comments and Suggestions for Authors

Authors investigate digit span tests (forward and backward) and single digit modality test to compare if the first ones are more sensitive for multiple sclerosis detection than the latter one and if they correlate with different brain regions. The manuscript is clearly written, however there are few concerns that I would like to address.

  1. Every table should be better described. It is not clear how to read and where to focus in the tables in order to observe differences between the groups and different conditions.
  2. Why Table 2 and Table 3 have first row bolded? Seems that the results in the first row are as significant as in the rest of rows of these two tables. This should be reflected in the table legend.
  3. Methods section is missing Random Forest description, how it was performed, did Authors provide both groups to the RF with different conditions together or each group with different conditions separately?
  4. Figure 1 needs a better legend, at the moment only the abbreviations are given.
  5. Same for Figure 2.
  6. Authors claim that DSB was found as the most sensitive to identify the MS. This result is not evident neither in the figures nor in the tables. Is it possible to separately show the differences of the three tests for the two groups?

Author Response

  1. Every table should be better described. It is not clear how to read and where to focus in the tables in order to observe differences between the groups and different conditions. Thank you very much for the advise. It is corrected.
  2. Why Table 2 and Table 3 have first row bolded? Seems that the results in the first row are as significant as in the rest of rows of these two tables. This should be reflected in the table legend. Thank you very much for the advise. It is corrected.
  3. Methods section is missing Random Forest description, how it was performed, did Authors provide both groups to the RF with different conditions together or each group with different conditions separately? We agree, RF description is added, thank you
  4. Figure 1 needs a better legend, at the moment only the abbreviations are given. The corrected Legend is included.
  5. Same for Figure 2. The corrected Legend is included.

  1. Authors claim that DSB was found as the most sensitive to identify the MS. This result is not evident neither in the figures nor in the tables. Is it possible to separately show the differences of the three tests for the two groups? The explanation of the results is included.

Reviewer 2 Report

Comments and Suggestions for Authors

Thank you for submitting this very interesting ans carefully performed study. 

DSMT, DSB and DSP are three interesting screening tools, and DSB/DSP add exploration of attention and working memory. Do MSB/MNP require more time? Do the testees find it demanding? Have they been tested in large populations? 

Judging from references, your MRS data are novel and pave the way for similar further work. 

It woukd be interesting to correlate the findings with biomarkers such as NFl and GFAP.

The MRS findings are clear, for this small and selected cohort. 

Is there a difference between women and men? The results would probably be different in more severe MSp. 

Please explain why you opted for brain regions devoid of macroscopic lesions. 

Do you have tractography data, in view of the importance association pathways?

Comments on the Quality of English Language

There a very few edits to  do, such as MS rather than SM near the end of the text. 

Author Response

DSMT, DSB and DSP are three interesting screening tools, and DSB/DSP add exploration of attention and working memory. Do MSB/MNP require more time? Do the testees find it demanding? Have they been tested in large populations? 

Thank you for the question. Methodology was extended and SDMT,DSB/DSF tests were explained. DSB/DSF took only several minutes and some of our patients found DSB more  difficult than SDMT. The tests can be used in general population.

Judging from references, your MRS data are novel and pave the way for similar further work. Thank you

It would be interesting to correlate the findings with biomarkers such as NFl and GFAP. An excellent suggestion, thank you very much or the interesting question. This note was included to the manuscript.

            The MRS findings are clear, for this small and selected cohort. Thank you

Is there a difference between women and men? The results would probably be different in more severe MSp. Diffferences between women and men were not significant.

          Please explain why you opted for brain regions devoid of macroscopic lesions. 

          We chose brain regions that have connections to pathways responsible for cognition  

          and memory. We examined metabolic changes of parts of GM and WM independent of

          the presence of demyelinating lesions.

Do you have tractography data, in view of the importance association pathways? Tractography is another method witch would be used in future, thank you for your suggestion.

Comments on the Quality of English Language

There a very few edits to  do, such as MS rather than SM near the end of the text. 

Thank you

Round 2

Reviewer 1 Report

Comments and Suggestions for Authors

The manuscript is improved. It can be accepted for publication.